# Methods and results used in the development of a consensus-driven extension to the Consolidated Standards of Reporting Trials (CONSORT) statement for trials conducted using cohorts and routinely collected data (CONSORT-ROUTINE)

Mahrukh Imran,[1] Linda Kwakkenbos,[2] Stephen J McCall,[3,4,5] Kimberly A McCord,[6] Ole Fröbert [6],[7] Lars G Hemkens,[6] Merrick Zwarenstein,[8,9] Clare Relton,[10] Danielle B Rice [6],[1,11] Sinéad M Langan [6],[12] Eric I Benchimol,[13,14,15] Lehana Thabane,[16] Marion K Campbell,[17] Margaret Sampson,[18] David Erlinge,[19] Helena M Verkooijen,[20,21] David Moher [6],[22] Isabelle Boutron,[23,24,25] Philippe Ravaud,[23,24,25] Jon Nicholl,[26] Rudolf Uher,[27] Maureen Sauvé,[28,29] John Fletcher,[30] David Torgerson,[31] Chris Gale [6],[32] Edmund Juszczak,[3,33] Brett D Thombs [6] [1,34]

► Prepublication history and supplemental material for this paper is available online. To view these files, please visit the journal online (http://dx.doi.org/10.1136/bmjopen-2021-049093).

For numbered affiliations see end of article.

**Correspondence to**
Dr Brett D Thombs;
brett.thombs@mcgill.ca

## ABSTRACT

**Objectives** Randomised controlled trials conducted using cohorts and routinely collected data, including registries, electronic health records and administrative databases, are increasingly used in healthcare intervention research. A Consolidated Standards of Reporting Trials (CONSORT) statement extension for trials conducted using cohorts and routinely collected data (CONSORT-ROUTINE) has been developed with the goal of improving reporting quality. This article describes the processes and methods used to develop the extension and decisions made to arrive at the final checklist.

**Methods** The development process involved five stages: (1) identification of the need for a reporting guideline and project launch; (2) conduct of a scoping review to identify possible modifications to CONSORT 2010 checklist items and possible new extension items; (3) a three-round modified Delphi study involving key stakeholders to gather feedback on the checklist; (4) a consensus meeting to finalise items to be included in the extension, followed by stakeholder piloting of the checklist; and (5) publication, dissemination and implementation of the final checklist.

**Results** 27 items were initially developed and rated in Delphi round 1, 13 items were rated in round 2 and 11 items were rated in round 3. Response rates for the Delphi study were 92 of 125 (74%) invited participants in round 1, 77 of 92 (84%) round 1 completers in round 2 and 62 of 77 (81%) round 2 completers in round 3. Twenty-seven members of the project team representing a variety of stakeholder groups attended the in-person consensus meeting. The final checklist includes five new items and eight modified items. The extension Explanation & Elaboration document further clarifies aspects that are important to report.

**Conclusion** Uptake of CONSORT-ROUTINE and accompanying Explanation & Elaboration document will improve conduct

## Strengths and limitations of this study

► We followed a five-step process to develop Consolidated Standards of Reporting Trials statement extension for trials conducted using cohorts and routinely collected data (CONSORT-ROUTINE), consistent with Enhancing the QUAlity and Transparency Of health Research guidance.

► Items were informed by reporting guidelines on similar research designs, a scoping review, a three-round Delphi process and expert members of the guideline development team.

► CONSORT-ROUTINE was reviewed and tested at various stages of the development by project team members and key stakeholders.

► The limited methodological literature on trials conducted using cohorts and routinely collected data was a limitation in developing the extension.

► Similar to other reporting guidelines, CONSORT-ROUTINE will require re-evaluation and revisions over time to ensure that it is kept up to date with evolving methodology and practice of trials using cohorts and routinely collected data.

of trials, as well as the transparency and completeness of reporting of trials conducted using cohorts and routinely collected data.

## BACKGROUND

The use of reporting guidelines, including the Consolidated Standards of Reporting Trials (CONSORT) statement, improves the transparency and completeness of reports of results from randomised controlled trials (RCTs).[1–4] The CONSORT statement helps to facilitate critical appraisal and interpretation of RCTs by providing guidance to authors on a minimal set of items that should be reported for all trials.[5] The CONSORT 2010 guideline aimed to improve the reporting of two-arm parallel group RCTs. Extensions of the CONSORT statement have been developed to encourage better reporting of other trial designs, including, for instance, multiarm parallel group randomised trials, cluster trials, pilot and feasibility trials and pragmatic trials.[6–9]

There is a growing interest in RCTs conducted using cohorts or routinely collected data, including registries, electronic health records (EHRs) and administrative databases.[10–14] In a cohort, a group of individuals is gathered for the purpose of conducting research, whereas routinely collected data refer to data initially collected for purposes other than research or without specific a priori research questions developed before collection.[15 16] Trials may use a cohort or routinely collected data for: (1) identification of eligible participants, (2) outcome ascertainment and (3) to implement an intervention, or for a combination of these purposes. For example, in registry-based RCTs, a registry could be used to identify eligible participants for a trial, for the collection of participant baseline characteristics and as the source of outcome data; some registries have used interactive technology to actively flag participants for RCT enrolment as patient data are entered into the registry.[12] In some EHR trials, the EHR itself is used to implement an intervention. For example, one RCT tested an intervention to reduce antibiotic prescribing by feeding back personalised antibiotic prescription data to primary care physicians.[17]

The use of cohorts and routinely collected data may make RCTs easier and more feasible to perform by reducing cost, time and other resources.[18 19] It may also facilitate the conduct of trials that more closely replicate real-world clinical practice. These trial designs, however, are relatively recent innovations, and published RCT reports may not describe important aspects of their methodology in a standardised way. Trials conducted using cohorts and routinely collected data share certain elements with conventional RCTs, but there are also distinctive elements to report that are not covered in the CONSORT 2010 statement. The REporting of studies Conducted using Observational Routinely-collected Data (RECORD) statement provides guidance on reporting of studies conducted using routinely collected data but does not address RCT-specific methodological and reporting considerations.[20] Research conducted using routinely collected data presents unique methodological challenges that are often insufficiently reported, but there is scant guidance on methods and reporting of trials conducted using routinely collected data or cohorts.[21 22]

An extension to the CONSORT statement for RCTs conducted using cohorts and routinely collected data was developed using methods recommended for developing reporting guidelines.[23] This article describes, in detail, the consensus-based development process. The main aims of this article are to: (1) describe the methods and processes used in the development of the CONSORT Extension for Trials Conducted Using Cohorts and Routinely Collected Data (CONSORT-ROUTINE)[24] and (2) describe decisions made to arrive at the final checklist and the accompanying Explanation & Elaboration statement.

## METHODS

The project was registered with the Enhancing the QUAlity and Transparency Of health Research (EQUATOR) network.[25] We followed the EQUATOR network's guidelines for recommended methods and processes for developing, disseminating and implementing healthcare reporting guidelines.[23] These methods have been used in the development of other similar EQUATOR guidelines. Figure 1 illustrates the five parts of the development process for this guideline.

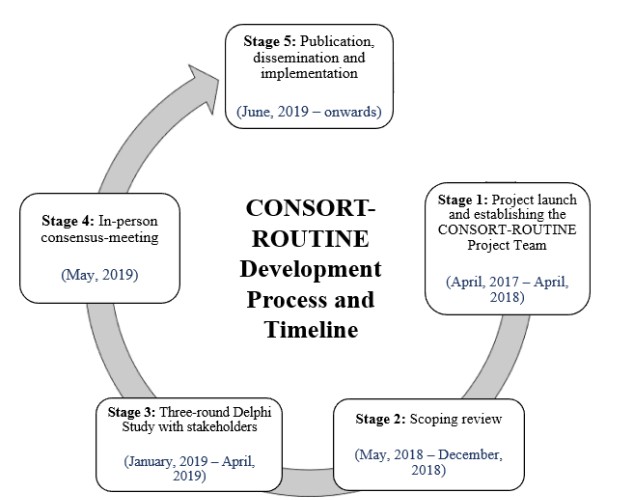

**Figure 1** Development process of the CONSORT Extension for Trials Conducted Using Cohorts and routinely Collected Data (CONSORT-ROUTINE).

## Project phase 1: project launch, establishment of team and funding

### Need for the guideline and literature review

An initial informal review of reports of published protocols and reports of trials using cohorts and routinely collected data by BDT and LK suggested that there appeared to be deficiencies in reporting of such trials. For instance, many reports did not adequately describe the cohort or database from which trial participants were recruited, processes used to link participants across databases were not always provided and it was sometimes unclear whether trial outcomes were assessed by the triallists or ascertained via existing databases used to conduct the trial. A review of the EQUATOR website and published literature indicated that there was no existing reporting guideline for these types of trials. The RECORD statement addresses reporting issues related to routinely collected data but does not include guidance on reporting of trials. Many trials conducted using routinely collected data are pragmatic or use cluster designs, for instance, but CONSORT extensions for those types of trials do not address issues germane to the use of cohorts or routinely collected data to conduct trials.[7 9]

### Project launch and identification of CONSORT-ROUTINE project members

Initial discussions on developing a CONSORT extension for RCTs conducted using cohorts occurred in November 2016 at the Trials within Cohorts symposium in London, UK (LK, MZ, CR and BDT).[26] Discussions continued virtually and key people involved in cohort-embedded trials or the EQUATOR network were approached during December 2016 (HMV, DM, IB, PR, JN, RU and DT). It was suggested that trials conducted in registries had many characteristics similar to those in cohorts, and there was agreement to include registry-based trials in the extension. People with expertise in registry-based trials were approached in March 2017 (OF, LT, MKC and DE), and an experienced librarian (MSam) and patient representative familiar with trials conducted using cohorts (MSau) were also included in the group at that point.

The project was registered on the EQUATOR website in April 2017. During the preparatory phase, while developing searches and reviewing example publications, we became aware that trials conducted using EHRs and administrative databases also shared similar characteristics with trials in cohorts and registries, and it was decided to expand the scope to trials conducted using cohorts and routinely collected data. In July 2017, triallists, who were leading the development of a reporting guideline for EHRs, joined the project group (EJ and CG). Given the relevance of their previous work and their expertise (LH, SL, DM and EIB), authors who had been involved in the development of the RECORD statement were invited to join the team.[20] Several doctoral students also joined

the project team (SJM, KAM and DBR). A steering committee comprising of 10 members with key expertise for consultation was established. A research coordinator (MI) was hired in April 2018 to manage the project, and an experienced journal editor was invited to join (JF). The group communicated regularly throughout the process via a number of virtual meetings, using an online platform to conduct teleconferences, as well as through email discussions.

### Rationale for developing one checklist versus four different checklists for trials conducted using cohorts, registries, EHRs and administrative databases

Team members discussed the advantages and disadvantages of creating individual checklists for each of the four types of data versus a single checklist for all four. It was determined that, although there are some differences in the implementation of trials across the different types of data sources, the methodological principles are similar, and there is substantial overlap in the design, conduct and factors that may influence interpretability. Thus, the steering committee reached consensus to develop a single statement, addressing any differences by including 'if applicable' to items in the checklist that may not apply to all trial designs and to clarify differences in the Explanation & Elaboration publication as deemed necessary.

### Funding

The project team obtained its main source of funding from a grant from the Canadian Institutes of Health Research Institutes (CIHR) to support the development of the guideline (BDT, OF, EJ, LK, CR; Grant #PJT-156172). EJ and CG also obtained funding from the UK National Institute of Health Research Clinical Trials Unit Support Funding - Supporting efficient/innovative delivery of NIHR research. In addition, funding to hold the face-to-face meeting was provided by a Planning and Dissemination Grant from CIHR (BDT and LK; Grant #PCS - 161863) and by contributions from Queen Mary University of London, the University of Sheffield, McGill University and the Lady Davis Institute for Medical Research of the Jewish General Hospital in Montreal, Canada.

A project protocol was developed and published.[22]

## Project phase 2: scoping review

A preliminary 'long list' of possible reporting items was formulated by LK and KAM based on review of the CONSORT 2010 statement items, the Strengthening the Reporting of Observational Studies in Epidemiology (STROBE)[27] and the RECORD statements,[20] as well as discussions with steering committee members. The STROBE and RECORD statements were considered the most relevant to this project because of their focus on reporting of observational studies and non-interventional studies using routinely collected data.

A scoping review was conducted to identify: (1) articles on the methodology or reporting of RCTs conducted using cohorts or routinely collected data that could

inform the development of new items or modification of existing CONSORT items; and (2) trial reports to identify aspects of reporting that need improvement and examples of good reporting of potential checklist items that could be used to support CONSORT-ROUTINE.[28] We searched for relevant articles on trials conducted using cohorts, registries, EHRs and administrative databases from 2007 to 2018. After screening articles for inclusion and exclusion at the abstract and full-text level, 10 people from the team independently reviewed the included papers and provided suggestions for modifications or additional reporting guideline items until no new ideas emerged (saturation). Suggestions were added in a standardised, shared spreadsheet. At the same time, team members provided examples of good reporting for each proposed item or item modification. Additionally, the review helped us to create a list of authors with experience in these trial designs as potential participants for the Delphi study. Search terms used in the scoping review are shown in online supplemental file 1.

### Project phase 3: Delphi study

The objectives of our Delphi study were: (A) to obtain feedback on the importance of including each candidate item in CONSORT-ROUTINE; (B) to improve the wording of items considered important; and (C) to elicit suggestions for additional items not in the existing list. We aimed to engage key stakeholders across different sectors and backgrounds. There are not fixed guidelines on the sample size of Delphi studies, and the ideal number of participants may depend on the complexity of the topic, the likely heterogeneity of relevant experiences and viewpoints, and resources available to manage the data generated.[29–31] Many studies use small groups of experts (eg, <20), but we believed that a larger group with diverse expertise would best complement the knowledge of the project team. Thus, we sent out an invitation to reporting guideline developers (including those involved in previous CONSORT extensions), funders, journal editors, patient representatives, trial methodologists, epidemiologists, meta-research authors, ethicists, biostatisticians and clinical triallists who were identified by members of the project team. We also encouraged recipients of the invitation to forward the invitation to other potentially interested stakeholders.

The Delphi surveys were built and hosted using an online survey platform in Qualtrics. During registration, we gathered demographic and professional background characteristics of participants, including geographical location, self-identified stakeholder group (eg, clinical trials user, clinical triallist and methodologist), employment sector, years of experience in trials research and research experience in trials conducted using cohorts or routinely collected data.

Registered participants received a link to access each of the three rounds of the Delphi survey. In each round, we asked participants to rate their perceptions about the importance of each suggested reporting item by ranking

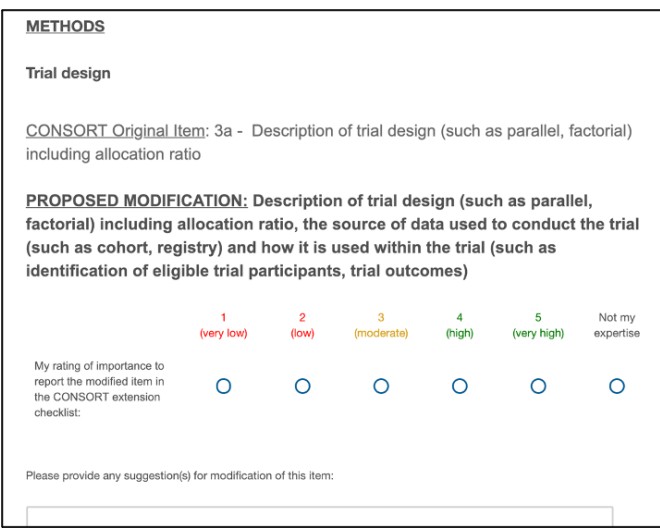

**Figure 2** Example of a round 1 Delphi survey item as presented in the online survey. CONSORT, Consolidated Standards of Reporting Trials.

items based on how essential they are for reporting on a 1–5 Likert scale (1=not essential; 5=essential). There is not consensus on the ideal number of Likert categories or groupings for decision-making, but it is common to use between 4-point and 7-point scales.[30]

Responses were categorised as follows:

1–2=low score (item should not be part of CONSORT-ROUTINE checklist).

3=moderate (item should be discussed).

4–5=high score (item should be part of CONSORT-ROUTINE checklist).

Participants also had the option to select 'Not my expertise' for items if they believed that they did not have the appropriate level of expertise to rate an item. Figure 2 shows a screenshot of an example proposed modification item from the survey:

Items from the CONSORT 2010 statement for which modifications were initially not proposed were also included in the survey so that participants could provide comments or make recommendations for modifications to these items. For all items (proposed modifications and CONSORT 2010 items), we provided participants with the opportunity to give open-ended feedback, using free-text boxes provided at the bottom of each survey page and at the end of the survey. At the end of the survey, participants were asked to provide any additional items that they believed would be important for reporting in trials conducted using cohorts and routinely collected data but that had not been included in the proposed set of new and modified items.

We launched round 1 of the survey on 4 February 2019 with 2 weeks to provide responses. Round 2 was launched on 4 March 2019, and round 3 was launched on 1 April 2019. After each round, the Qualtrics built-in analysis software was used to generate a distribution of scores and to aggregate group results for each item (mean score, maximum and minimum score, SD, variance

and percentage ratings of 1–5 ranking for items) and summary statistics were circulated among all participants. Individual responses were not fed back. In addition, a bar chart with the ratings and counts for each item was created. Following each round of the survey, the CONSORT-ROUTINE steering committee members reviewed the survey results independently and then met via teleconference to discuss and analyse the results of the survey. During these meetings, decisions were made on how to address comments from participants by modifying, adding or combining items. Notes were also made on comments that reflected a need for explanation in the Explanation & Elaboration companion to the checklist.

We predefined consensus as at least 2/3 of responders rating the importance of an item as 'high' or 'very high'. Items that reached consensus for inclusion were not rated again in the next round. For some items that did not reach consensus, the wording of items was revised based on participants' suggestions. Items that did not reach consensus were rated again in the next round in their original or revised form. Reports summarising the Delphi results were circulated after each round including summary statistics such as counts, means, SD and variances for the responses on each item. Reminder emails were sent 1 week prior to the deadline and extensions were provided if requested for all three rounds in order to maximise participation.

Since the Delphi Study was advisory, all items were reviewed and vetted again at the in-person consensus meeting, and comments provided by participants of the Delphi Study were taken into consideration while making decisions to include or exclude items.

### Project phase 4: in-person consensus meeting and development of checklist publication

A 2-day in-person consensus meeting was held on 13–14 May 2019 in London, UK. The purpose of the meeting was to discuss the Delphi results, make decisions on items to retain in the final checklist, make any necessary modifications to items and suggest reporting aspects that should be addressed in the Explanation & Elaboration documentation supporting the checklist. The meeting was attended by 26 members of the CONSORT-ROUTINE Group.

We used approaches similar to those used in previous consensus meetings for other guidelines. Participants were provided with the results of the initial long-list generation and the Delphi study in advance of the meeting. At the meeting, steering committee members first presented the background and an update on work done to date, in order to facilitate the discussions. Session chairs then separately presented items from the preliminary checklist, results of the Delphi study and feedback from stakeholders, after which the group discussed in an open forum. Decisions were made on items to be modified or added based on the following criteria: (1) whether they addressed elements unique to trials conducted using cohorts or routinely collected data versus elements applicable to any trial and (2) whether

they reflected information that should be included in a minimum reporting set of items. Notes were taken, and the discussions were audio-recorded to ensure that the content was accurately captured.

Following the consensus meeting, refinement of the content and wording of the items was continued through online group discussions with CONSORT-ROUTINE project team members. The initial version of the checklist was pilot-tested by circulating it among stakeholders in order to assess its usability and to identify any challenges that might arise while applying the checklist. Pilot-testing the checklist also provided insight into issues that should be addressed in detail in the Explanation & Elaboration statement.

### Project phase 5: publication, dissemination and implementation

As with several previous CONSORT extensions, it was decided to publish the reporting checklist with a detailed Explanation & Elaboration statement in the same document.[6–9] The Explanation & Elaboration statement is intended to provide an in-depth explanation of the scientific rationale for each recommendation, together with an example of clear reporting for each item.

In addition to publication of the reporting guideline checklist and Explanation & Elaboration material, to attempt to maximise uptake, we will undertake additional dissemination activities, including presentations and workshops at conferences and other venues. We also plan to seek endorsement of the guideline by journal editors. Research has shown that formal endorsement and adoption of the CONSORT statement by journals is associated with improved quality of reporting.[2] Studies conducted by members of our team have benchmarked pre-extension reporting completeness in trials conducted in cohorts, registries, EHRs, and administrative databases.[32–34] There were not enough examples of completed cohort-embedded trials for benchmarking reporting.

The final CONSORT-ROUTINE checklist has been published.[24]

### Patient and public involvement

One of the members of our CONSORT-ROUTINE team, MSau, is a patient organisation leader. She has been involved in working with researchers to establish a cohort of patients living with the rare disease scleroderma, which supports RCTs of trials of online rehabilitation, self-management and psychological intervention programmes.

### RESULTS
### Stage 2: scoping review and initial long list of potential items
The scoping review sought methods articles and reports of trials conducted using cohorts, registries, EHRs or administrative databases.

### Cohorts

The database search identified 1185 publications, of which 1062 were excluded after title and abstract screening and 37 after full-text review. A total of 86 studies were included in the scoping review, including 15 papers on methodological considerations of using cohorts for conducting RCTs. All trials used the cohort for both identification of patients and outcome ascertainment.

### Registries

The search identified 234 publications, of which 143 received full-text review. A total of 106 publications were eligible, including 95 trial reports or protocols (both identification of patients and outcome ascertainment (n=27); identification of patients only (n=28); outcome ascertainment only (n=40)) and 11 papers on methodological considerations.

### Electronic health records

The search identified 2085 citations, of which 548 studies were reviewed at the full-text level. A total of 289 eligible publications, including 263 trial protocols or reports (both identification of patients and outcome ascertainment (n=169); identification of patients only (n=38); outcome ascertainment only (n=56)) and 26 articles that described methodological considerations.

### Administrative databases

The search identified 663 citations, of which 151 full texts were reviewed. There were a total of 117 trial protocols or reports included (both identification of patients and outcome ascertainment (n=57); identification of patients only (n=1); outcome ascertainment only (n=58)) and one paper on methodological considerations.

### Delphi study results

Of 125 people invited to take part in the Delphi study, 115 people registered via an online survey, and 92 (74%) provided responses on the items in round 1. Figures 3 and 4 present the types of stakeholder groups that completed round 1 of the Delphi study and the type of trials conducted using cohorts or routinely collected databases with which they had familiarity. Participants belonging to more than one category had the option of checking multiple options in the survey.

### Round 1

Of the 92 participants who completed the round 1 survey, 90 provided valid ratings and two provided comments but not ratings. Of the 27 items rated in round 1, 14 reached consensus to be included in discussions at the consensus meeting; the other 13 did not reach consensus and were included in round 2. Based on round 1 feedback, a total of 11 items were modified for review in round 2, including

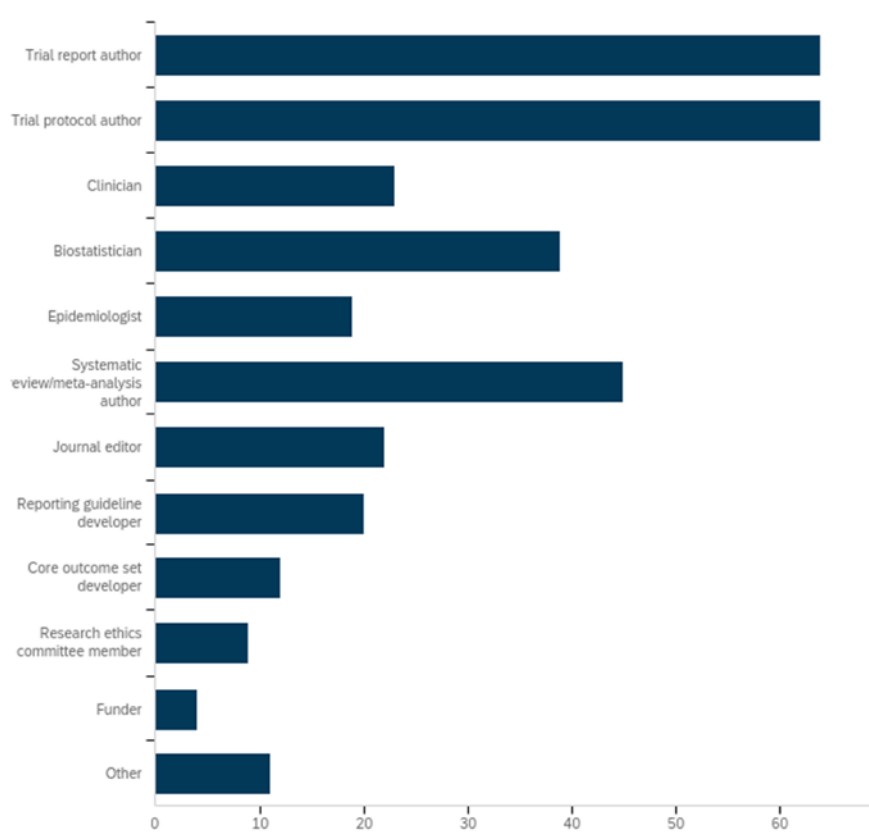

**Figure 3** Professional roles reported by participants who completed round 1 of the CONSORT-ROUTINE Delphi study (%). Participants could report more than one role. CONSORT-ROUTINE, Consolidated Standards of Reporting Trials Extension for Trials Conducted Using Cohorts and Routinely Collected Data.

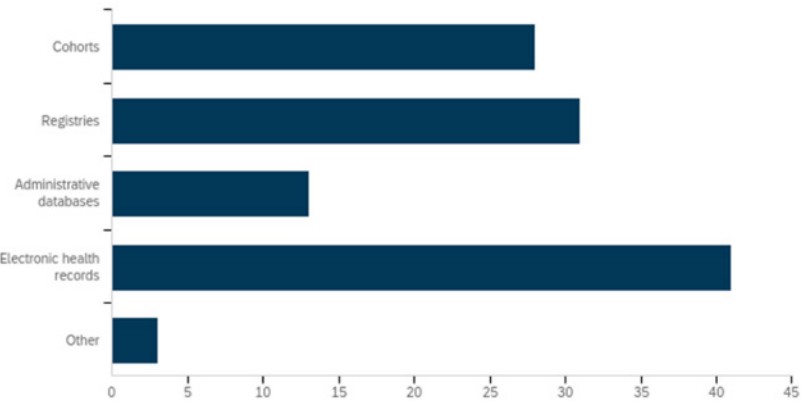

**Figure 4** Participants of round 1 of the CONSORT-ROUTINE Delphi study by type of cohort or routinely collected database with which they had familiarity (%). Participants could report more than one. CONSORT-ROUTINE, Consolidated Standards of Reporting Trials Extension for Trials Conducted Using Cohorts and Routinely Collected Data.

two items that were combined into one. No items were excluded from the checklist.

### Round 2

Of the 92 participants who completed round 1, 77 (84%) completed the round 2 survey. Of the 13 items rated, 2 reached consensus for inclusion in consensus meeting discussions, and 11 did not reach consensus in round 2. Based on round 2 feedback, eight items were modified prior to round 3.

### Round 3

Of the 77 people who completed round 2, 62 (81%) completed round 3. Of the 11 items in round 3, five items reached consensus in round 3. The remaining six items did not reach consensus after the three rounds.

There were several new items suggested via the Delphi process but not added to the potential item list. The main reasons why some items were suggested but not incorporated were:

1. The suggestion was encapsulated in CONSORT 2010 items, was already captured by proposed new or modified items or could be captured by further modifying new or modified items.
2. The suggestion was not specific to trials conducted using cohorts and routinely collected data and, thus, was recommending a change to the CONSORT 2010 checklist, which was not the task of the CONSORT-ROUTINE group.

Summary results of the three rounds can be accessed at: https://osf.io/4zh6f/

### In-person consensus meeting

Table 1 summarises the CONSORT-ROUTINE group's discussions and advisory decisions for each of the items that was discussed during the in-person meeting. If there were differing opinions on the inclusion or exclusion of items and consensus could not be reached, voting was implemented by the session chair, with an 80% threshold for inclusion in the checklist as part of the minimal set

of recommended reporting items. The key recommendations that emerged were as follows:

► Proposed modification to CONSORT 2010 items: it was recommended to retain proposed modifications to seven CONSORT 2010 items. These modifications pertained to differences in mechanisms used to conduct trials using cohorts or routinely collected databases. As in previous CONSORT extensions, some of the recommended changes end with 'if applicable' to show that some information which authors are being asked to report might not be relevant or applicable for their particular RCT or the particular type of data that was used in the RCT.

► Proposed additional items: consensus was reached to include six additional items and to add a new subheading, 'Cohort or routinely collected database', to the checklist.

A recurrent discussion point was the need to minimise adding new items to the abstract unless they are essential due to word limits imposed by journals. A suggestion was made to expand the explanatory text of the Explanation & Elaboration document for nine unchanged CONSORT 2010 items to clarify additional requirements for reporting aspects of the trial without modifying the item: item 1a (identification as a randomised trial in the title), item 4b (settings and location where the data were collected), item 5 (intervention), item 13b (losses and exclusions after randomisation), item 14a (dates of recruitment/follow-up), item 15 (baseline data), item 20 (limitations), item 21 (generalisability) and item 24 (study protocol). For the abstract, there was an agreement to include an additional item to the abstract for naming the cohort or routinely collected database (item 1c). This item was later merged with item 1b from the CONSORT 2010 checklist after discussion with the project team (table 1). Thus, the final extension checklist included eight modified items and five new items.[24]

### CONSORT-ROUTINE pilot test

The preliminary version of the checklist was pilot-tested by 17 people who had been previously involved in conducting trials using cohorts and routinely collected data. Based on feedback received from the pilot test, there were minor

**Table 1** Consensus meeting discussions and advisory decisions for the checklist items

| Section/topic | CONSORT 2010 item | CONSORT ext. item | CONSORT 2010 item | Suggested modified or additional extension items | Consensus status (Delphi) | Summary of the discussion, decisions and suggestions made during the CONSORT-ROUTINE in-person consensus meeting | Final checklist item to be included in CONSORT-ROUTINE |
|---|---|---|---|---|---|---|---|
| **Title and abstract** | | | | | | | |
| | 1a | 1a | Identification as a randomised trial in the title. | Identification as a randomised trial in the title, including that it was a trial conducted using a cohort or routinely collected source of data (modified). | Not reached. | Discussed the need for a modification to the original item. It was noted that multiple databases or types of databases could be used to conduct a trial and stating all would not be feasible as journals might have title length restrictions. Decision: do not include the modification and retain the CONSORT 2010 item; expand the E&E text for clarification. | Identification as a randomised trial in the title. |
| | 1b | 1b | Structured summary of trial design, methods, results and conclusions (for specific guidance see CONSORT for abstracts). | | | No suggested modification. Decision: retain the CONSORT 2010 item. Note: additional item 1c was later merged with this item (see further). | Structured summary of trial design, methods, results and conclusions (for specific guidance, see CONSORT for abstracts). Specify that a cohort or routinely collected data were used to conduct the trial and, if applicable, provide the name of the cohort or routinely collected database(s). |
| | | | | The source(s) of data used to conduct the trial should be specified in the abstract (additional). | Reached for inclusion. | Noted the importance of stating the cohort or routinely collected database(s) used to conduct the trial in the abstract, if not in the title. Decision: include the suggested new item with revisions. Note: the item was later merged with item 1b from CONSORT 2010. | |
| | | | | If linkage between multiple sources of data was conducted for the study, this should be clearly stated in the abstract (additional). | Not reached. | Mixed views on the necessity of reporting the suggested new item in the abstract. Agreed that linkage is important to report in the body of the paper but not necessarily the abstract. Decision: do not include the suggested new item. | |
| | | | | The proportion of participants offered and the proportion that accepted the intervention should be reported (for trials conducted using the cohort multiple RCT design) (additional). | Reached for inclusion. | Mixed views on the necessity of reporting the suggested new item in the abstract. Agreement that the information is important to report but not essential for the abstract due to word count restrictions. In addition, this applies to one trial design used in cohorts but not all cohort trials and not trials using other types of data. The item was merged with CONSORT 2010 item 13a (14a in the final extension checklist) pertaining to participant flow. Decision: do not include the suggested new item in the abstract but include in item 14a in the final extension checklist. | |

Continued

**Table 1** Continued

| Section/topic | CONSORT 2010 item | CONSORT ext. item | CONSORT 2010 item | Suggested modified or additional extension items | Consensus status (Delphi) | Summary of the discussion, decisions and suggestions made during the CONSORT-ROUTINE in-person consensus meeting | Final checklist item to be included in CONSORT-ROUTINE |
|---|---|---|---|---|---|---|---|
| **Introduction** | | | | | | | |
| Background and objectives | 2a | 2a | Scientific background and explanation of rationale. | | | Discussed the importance of reporting the rationale for conducting the trial using a cohort or routinely collected database but decided against modifying original CONSORT 2010 item. Decision: retain the CONSORT 2010 item. | Scientific background and explanation of rationale. |
| | 2b | 2b | Specific objectives or hypotheses | Specific objectives or hypotheses | | No suggested modification. Decision: retain the CONSORT 2010 item. | Specific objectives or hypotheses. |
| **Methods** | | | | | | | |
| Trial design | 3a | 3a | Description of trial design (such as parallel, factorial) including allocation ratio. | Description of trial design (such as parallel and factorial) including allocation ratio, the source(s) of data used to conduct the trial (such as cohort and registry) and how the data are used within the trial (such as identification of eligible trial participants, trial outcomes) (modified). | | Noted that key elements of the study design and cohort or database(s) used for the trial should be stated early in the methods section, as well as the extent to which the database was used in the trial. Decision: include the modified item. | Description of trial design (such as parallel and factorial) including allocation ratio, that a cohort or routinely collected database(s) used to conduct the trial (such as electronic health record and registry) and how the data were used within the trial (such as identification of eligible trial participants and trial outcomes). |
| | 3b | 3b | Important changes to methods after trial commencement (such as eligibility criteria), with reasons. | Important changes to methods after trial commencement (such as eligibility criteria), with reasons. | | No suggested modification. Decision: retain the CONSORT 2010 item. | Important changes to methods after trial commencement (such as eligibility criteria), with reasons. |
| Cohort or routinely collected database (additional header) | | ROUTINE-1 | | Description of the source(s) of data used to conduct the trial, including the setting, locations, relevant dates, periods of recruitment, follow-up and data collection (additional). | Reached for inclusion | Agreed on the importance of reporting the item. Decision: include the suggested new item. | Name, if applicable, and description of the cohort or routinely collected database(s) used to conduct the trial, including information on the setting (such as primary care), locations and dates (such as periods of recruitment, follow-up and data collection). |
| | | | | Describe indicators of the quality of the source(s) of data used to conduct the trial including what types of quality checks have been performed and the entity responsible for the data (additional). | Reached for inclusion. | Mixed views on the necessity of the suggested new item. There were concerns that 'quality' is vague and the term 'accuracy and completeness' may better clarify the intent of the item. It was acknowledged that the accuracy and completeness of the cohort or database is important to report while (1) selecting participants and (2) ascertaining outcomes. Decision: do not include the suggested new item as a standalone item. The item was merged with extension items 5a and 7b (pertaining to participant selection and outcome ascertainment) in the finalised checklist. | |

Continued

**Table 1** Continued

| Section/topic | CONSORT 2010 item | CONSORT ext. item | Suggested modified or additional extension items | Consensus status (Delphi) | Summary of the discussion, decisions and suggestions made during the CONSORT-ROUTINE in-person consensus meeting | Final checklist item to be included in CONSORT-ROUTINE |
|---|---|---|---|---|---|---|
| | | | Describe modifications to the data collected in the source(s) of data used to conduct the trial, such as adding data items, if applicable (additional). | Reached for inclusion | Agreed that the suggested item is not necessarily unique to trials conducted using cohorts and routinely collected data. Decision: do not include the suggested new item; expand the E&E text for clarification. | |
| | | | Describe additional sources of data used to conduct the trial, if any (additional). | Reached for inclusion. | Mixed views on the necessity of the suggested new item as it is not unique to trials conducted using cohorts and routinely collected data. Decision: do not include the suggested new item; expand the E&E text for clarification. | |
| | | ROUTINE-2 | Give the eligibility criteria, the sources and methods of selection of participants, and methods of follow-up (for trials conducted using cohorts or registries) (additional). | Reached for inclusion | Discussed the importance of reporting the eligibility criteria for inclusion in the cohort or routinely collected database(s), but there was concern that elements related to follow-up are not specific to trials conducted using cohorts and routinely collected data. Decision: include the suggested new item with revisions; expand the E&E text for clarification of other aspects. | Eligibility criteria for participants in the cohort or routinely collected database(s). |
| | | ROUTINE-3 | Detail any use of record linkage across sources of data, the methods of linkage and methods of quality evaluation, if applicable (additional). | Reached for inclusion. | Suggestion to integrate wording from RECORD checklist for clarity. Decision: include the suggested new item adapted from RECORD. | State whether the study included person-level, institutional-level or other data linkage across two or more databases and, if so, linkage techniques and methods used to evaluate completeness and accuracy of linkage. |
| | | | Describe if (and how) participants were informed about the potential use of their data in randomised trials (additional). | Not reached | Mixed views on the necessity of the item as some believed that ethics considerations are beyond the scope of CONSORT, and ethics does not appear in CONSORT 2010. The group agreed to include the item as consent is an important issue with unique aspects in these trials, but that this should be presented as part of trial participants section. Decision: include the suggested new item with revisions and move to section 'Trial participants' as item 5c. | |

**Table 1** Continued

| Section/topic | CONSORT 2010 item | CONSORT ext. item | CONSORT 2010 item | Suggested modified or additional extension items | Consensus status (Delphi) | Summary of the discussion, decisions and suggestions made during the CONSORT-ROUTINE in-person consensus meeting | Final checklist item to be included in CONSORT-ROUTINE |
|---|---|---|---|---|---|---|---|
| Trial participants (modified header) | 4a | 4a | Eligibility criteria for participants. | Eligibility criteria for trial participants (modified). | | Agreed to merge with suggested new item (see next row). Decision: merge with suggested new item, 'Provide details of how eligible clusters/participants were identified from the source(s) of data used to conduct the trial'. | Eligibility criteria for trial participants, including information on how to access the list of codes and algorithms used to identify eligible participants, information on accuracy and completeness of data used to ascertain eligibility and methods used to validate accuracy and completeness (eg, monitoring, adjudication), if applicable. |
| | | | | Provide details of how eligible clusters/participants were identified from the source(s) of data used to conduct the trial (additional). | Reached for inclusion. | Suggested merging with CONSORT 2010 item 4a (5a in final checklist) and address accuracy and completeness of data. Decision: merge the suggested new item with CONOSRT 2010 item 4a (5a in the final checklist). | |
| | 4b | 4b | Settings and locations where the data were collected. | Settings and locations where the trial data were collected (modified). | Reached for inclusion. | The word 'trial' was dropped as the header 'Trial participants' clarifies the intent of the item. Decision: retain the CONSORT 2010 item; expand the E&E text for clarification. | Settings and locations where the data were collected. |
| | | ROUTINE-4 | | Details of information provided to participants from the source(s) of data who are selected for recruitment or inclusion in the trial, including any differences in information provided across trial arms (additional). | Not reached. | Extended discussions on the importance of the item as it might only be applicable to the controlled multiple RCT design. Agreement to formulate as a general item on consent as item 5c. Decision: do not include the suggested new item. The consent item was simplified and moved to this section. Expand the E&E text for clarification of consent issues. | Describe whether and how consent was obtained. |
| Interventions | 5 | 5 | The interventions for each group with sufficient details to allow replication, including how and when they were actually administered. | | | No suggested modification. Decision: retain the CONSORT 2010 item; expand the E&E text for clarification. | The interventions for each group with sufficient details to allow replication, including how and when they were actually administered. |
| | | | | Describe how the source(s) of data was used to implement the intervention, if applicable (eg, for trials conducted using electronic health records) (additional). | Reached for inclusion. | Debated the necessity of the new item as it is only applicable to trials conducted using electronic health records that may be used as intervention tools. Decision: do not include the suggested new item; expand the E&E text for clarification. | |
| Outcomes | 6a | 6a | Completely defined prespecified primary and secondary outcome measures, including how and when they were assessed. | Completely defined prespecified primary and secondary outcome measures, including how and | | Suggestion to merge with proposed new item, 'Provide source(s) of data for each outcome' (see below). Decision: item merged with suggested new item and included in the final checklist. | Completely defined prespecified primary and secondary outcome measures, including how and when they were ascertained and the cohort or routinely collected database(s) used to ascertain each outcome. |

Continued

**Table 1** Continued

| Section/topic | CONSORT 2010 item | CONSORT ext. item | CONSORT 2010 item | Suggested modified or additional extension items | Consensus status (Delphi) | Summary of the discussion, decisions and suggestions made during the CONSORT-ROUTINE in-person consensus meeting | Final checklist item to be included in CONSORT-ROUTINE |
|---|---|---|---|---|---|---|---|
| | | | | Provide source(s) of data for each outcome (additional). | Reached for inclusion. | Suggestion to merge with CONSORT 2010 item 6a. Decision: item merged with CONSORT 2010 item 6a (7a in the final checklist). | |
| | | ROUTINE-5 | | Provide a list of codes and algorithms used to define (and/or derive) the outcomes as online supplemental information, including validation, if applicable (additional). | Not reached. | Acknowledged the importance of reporting the list of codes and algorithms for ascertaining outcomes along with the accuracy and completeness of data and validation. Decision: include the suggested new item with revisions. | Information on how to access the list of codes and algorithms used to define or derive the outcomes from the cohort or routinely collected database(s) used to conduct the trial, information on accuracy and completeness of outcome variables and methods used to validate accuracy and completeness (eg, monitoring, adjudication), if applicable. |
| | | | | Detail any adjudication or external validation of data items from the source(s) of data used to conduct the trial, if applicable (additional). | Reached for inclusion. | Acknowledged the importance of reporting the item. There was agreement that validation should be reported while selecting participants and ascertaining outcomes and included as part of items 5a and 7b of extension checklist. Decision: address elements of proposed item as part of items 5a and 7b in the final checklist. | |
| | 6b | 6b | Any changes to trial outcomes after the trial commenced, with reasons. | | | No suggested modification. Decision: retain the CONSORT 2010 item. | Any changes to trial outcomes after the trial commenced, with reasons. |
| Sample size | 7a | 7a | How sample size was determined. | | | No suggested modification. Decision: retain the CONSORT 2010 item. | How sample size was determined. |
| | 7b | 7b | When applicable, explanation of any interim analyses and stopping guidelines. | | | No suggested modification. Decision: retain the CONSORT 2010 item. | When applicable, explanation of any interim analyses and stopping guidelines. |
| **Randomisation** | | | | | | | |
| Sequence generation | 8a | 8a | Method used to generate the random allocation sequence. | | | No suggested modification. Decision: retain the CONSORT 2010 item; expand the E&E text for clarification. | Method used to generate the random allocation sequence. |
| | 8b | 8b | Type of randomisation; details of any restriction (such as blocking and block size). | | | No suggested modification. Decision: retain the CONSORT 2010 item. | Type of randomisation; details of any restriction (such as blocking and block size). |
| Allocation concealment mechanism | 9 | 9 | Mechanism used to implement the random allocation sequence (such as sequentially numbered containers), describing any steps taken to conceal the sequence until interventions were assigned. | Mechanism used to implement the random allocation sequence, describing any steps taken to conceal the sequence until interventions were assigned, such as using automated random sequence generation concealed within source(s) of data (modified). | Reached for inclusion. | Discussion to clarify wording of the item. Decision: include the modified item with revisions. | Mechanism used to implement the random allocation sequence (such as embedding an automated randomiser within the cohort or routinely collected database(s)), describing any steps taken to conceal the sequence until interventions were assigned. |

Continued

**Table 1** Continued

| Section/topic | CONSORT 2010 item | CONSORT ext. item | CONSORT 2010 item | Suggested modified or additional extension items | Consensus status (Delphi) | Summary of the discussion, decisions and suggestions made during the CONSORT-ROUTINE in-person consensus meeting | Final checklist item to be included in CONSORT-ROUTINE |
|---|---|---|---|---|---|---|---|
| Implementation | 10 | 10 | Who generated the random allocation sequence, who enrolled participants and who assigned participants to interventions. | | | No suggested modification. Decision: retain the CONSORT 2010 item. | Who generated the random allocation sequence, who enrolled participants and who assigned participants to interventions. |
| Blinding | 11a | 11a | If done, who was blinded after assignment to interventions (eg, participants, care providers, those assessing outcomes) and how. | | | No suggested modification. Decision: retain the CONSORT 2010 item. | If done, who was blinded after assignment to interventions (eg, participants, care providers, those assessing outcomes) and how. |
| | 11b | 11b | If relevant, description of the similarity of interventions. | | | No suggested modification. Decision: retain the CONSORT 2010 item. | If relevant, description of the similarity of interventions. |
| Statistical methods | 12a | 12a | Statistical methods used to compare groups for primary and secondary outcomes. | | | No suggested modification. Decision: retain the CONSORT 2010 item. | Statistical methods used to compare groups for primary and secondary outcomes. |
| | 12b | 12b | Methods for additional analyses, such as subgroup analyses and adjusted analyses. | | | No suggested modification. Decision: retain the CONSORT 2010 item. | Methods for additional analyses, such as subgroup analyses and adjusted analyses. |
| **Results** | | | | | | | |
| Participant flow (a diagram is strongly recommended) | 13a | 13a | For each group, the numbers of participants who were randomly assigned, received intended treatment and were analysed for the primary outcome. | Describe in detail the numbers of clusters/participants in the source(s) of data used to conduct the trial, number screened for eligibility, randomly assigned, offered and accepted interventions (eg, cohort multiple RCTs), received intended treatment and analysed for the primary outcome (modified). | Reached for inclusion. | Suggestion to form a committee to draft example flow diagram and oversee the E&E. Decision: include the modified item; committee to oversee the E&E development. | For each group, the number of participants in the cohort or routinely collected database(s) used to conduct the trial and the numbers screened for eligibility, randomly assigned, offered and accepted interventions (eg, cohort multiple RCTs), received intended treatment and analysed for the primary outcome. |
| | | | | Describe any linkage of multiple sources of data, including the number of clusters/participants successfully linked (additional). | Reached for inclusion. | Debated the necessity of the item as a stand-alone item as linkage was addressed in item 4c. Suggested to include the number of clusters/participants successfully linked as part of the flow diagram. Decision: do not include the suggested new item; expand the E&E text for clarification. | |
| | 13b | 13b | For each group, losses and exclusions after randomisation, together with reasons. | | | No suggested modification. Discussed that the item should be tied to data accuracy and completeness, and linkage. Decision: retain the CONSORT 2010 item; expand the E&E text for clarification. | For each group, losses and exclusions after randomisation, together with reasons. |

Continued

**Table 1** Continued

| Section/topic | CONSORT 2010 item | CONSORT ext. item | CONSORT 2010 item | Suggested modified or additional extension items | Consensus status (Delphi) | Summary of the discussion, decisions and suggestions made during the CONSORT-ROUTINE in-person consensus meeting | Final checklist item to be included in CONSORT-ROUTINE |
|---|---|---|---|---|---|---|---|
| Recruitment | 14a | 14a | Dates defining the periods of recruitment and follow-up. | | | No suggested modification. Decision: retain the CONSORT 2010 item. | Dates defining the periods of recruitment and follow-up. |
| | 14b | 14b | Why the trial ended or was stopped. | | | No suggested modification. Decision: retain the CONSORT 2010 item. | Why the trial ended or was stopped. |
| Baseline data | 15 | 15 | A table showing baseline demographic and clinical characteristics for each group. | | | No suggested modification. Decision: retain the CONSORT 2010 item. | A table showing baseline demographic and clinical characteristics for each group. |
| | | | | A table showing baseline demographic and clinical characteristics for eligible participants who participated in the trial and those who did not (additional). | Reached for inclusion. | Agreement to not include the suggested new item as a stand-alone item. The information should be reported if possible, but not necessary, and implications should be addressed as part of 'Generalisability' (item 21). Decision: do not include the suggested new item. | |
| Numbers analysed | 16 | 16 | For each group, number of participants (denominator) included in each analysis and whether the analysis was by original assigned groups. | | | No suggested modification. Decision: retain the CONSORT 2010 item. | For each group, number of participants (denominator) included in each analysis and whether the analysis was by original assigned groups. |
| Outcomes and estimation | 17a | 17a | For each primary and secondary outcome, results for each group, and the estimated effect size and its precision (such as 95% CI). | | | No suggested modification. Decision: retain the CONSORT 2010 item. | For each primary and secondary outcome, results for each group, and the estimated effect size and its precision (such as 95% CI). |
| | 17b | 17b | For binary outcomes, presentation of both absolute and relative effect sizes is recommended. | | | No suggested modification. Decision: retain the CONSORT 2010 item. | For binary outcomes, presentation of both absolute and relative effect sizes is recommended. |
| Ancillary analyses | 18 | 18 | Results of any other analyses performed, including subgroup analyses and adjusted analyses, distinguishing prespecified from exploratory. | | | No suggested modification. Decision: retain the CONSORT 2010 item. | Results of any other analyses performed, including subgroup analyses and adjusted analyses, distinguishing prespecified from exploratory. |
| | | | | If outcomes for eligible patients in the existing source(s) of data who were not included in the trial are known, they should be reported (additional). | Not reached. | Agreement to not include the suggested new item as a stand-alone item. The information should be reported if possible, but not necessary, and implications should be addressed as part of 'Generalisability'. Decision: do not include the suggested new item; expand the E&E text for clarification in the 'Generalisability' section. | |
| Harms | 19 | 19 | All important harms or unintended effects in each group (for specific guidance see CONSORT for harms). | | | No suggested modification. Decision: retain the CONSORT 2010 item. | All important harms or unintended effects in each group (for specific guidance see CONSORT for harms). |

Continued

**Table 1** Continued

| Section/topic | CONSORT 2010 item | CONSORT ext. item | CONSORT 2010 item | Suggested modified or additional extension items | Consensus status (Delphi) | Summary of the discussion, decisions and suggestions made during the CONSORT-ROUTINE in-person consensus meeting | Final checklist item to be included in CONSORT-ROUTINE |
|---|---|---|---|---|---|---|---|
| **Discussion** | | | | | | | |
| Limitations | 20 | 20 | Trial limitations, addressing sources of potential bias, imprecision, and, if relevant, multiplicity of analyses. | | | No suggested modification. Decision: retain the CONSORT 2010 item. | Trial limitations, addressing sources of potential bias, imprecision, and, if relevant, multiplicity of analyses. |
| | | | | Discuss the implications of using data that were not created or collected to answer the specific research question(s) (additional). | Reached for inclusion. | Discussed that using routinely collected data is not necessarily a limitation, and the content of this item should be addressed in the 'Interpretation' section. Decision: do not include the suggested new item; merge with CONSORT 2010 item 22 (23 in the final checklist); expand the E&E text for clarification in the 'Generalisability' section. | |
| Generalisability | 21 | 21 | Generalisability (external validity and applicability) of the trial findings. | | | No suggested modification. Agreement to elaborate on the representativeness of the cohort or routinely collected database(s) used for the trial, including issues related to characteristics of eligible cohort or database participants who do not agree to participate in trial. Decision: retain the CONSORT 2010 item; expand the E&E text for clarification. | Generalisability (external validity and applicability) of the trial findings. |
| Interpretation | 22 | 22 | Interpretation consistent with results, balancing benefits and harms and considering other relevant evidence. | | | Item merged with the proposed new item 'Discuss the implications of using data that were not created or collected to answer the specific research question(s)'. Decision: include the modified item. | Interpretation consistent with results, balancing benefits and harms and considering other relevant evidence, including the implications of using data that were not collected to answer the trial research questions. |
| **Other information** | | | | | | | |
| Registration | 23 | 23 | Registration number and name of trial registry. | | | No suggested modification. Decision: retain the CONSORT 2010 item. | Registration number and name of trial registry. |
| Protocol | 24 | 24 | Where the full trial protocol can be accessed, if available. | | | No suggested modification. Decision: retain the CONSORT 2010 item; expand the E&E text for clarification. | Where the full trial protocol can be accessed, if available. |
| Funding | 25 | 25 | Sources of funding and other support (such as supply of drugs), role of funders. | Sources of funding and other support for the trial and the existing source(s) of data, role of funders (modified). | Reached for inclusion. | Suggested minor revision to the item. Decision: include the modified item with revision. | Sources of funding and other support for both the trial and the cohort or routinely collected database(s), role of funders |

CONSORT, Consolidated Standards of Reporting Trials; E&E, Explanation & Elaboration; RCT, randomised controlled trial; ROUTINE, Extension for Trials Conducted Using Cohorts and Routinely Collected Data.

modifications made to the wording of two items for clarity (item 1b and item 9) in the final checklist.[24]

## DISCUSSION

We have developed a consensus-driven extension to the CONSORT 2010 Statement for RCTs conducted using cohorts and routinely collected data.[24] CONSORT-ROUTINE contains minimum reporting requirements with appropriate flexibility as described in the Explanation & Elaboration part of our checklist document. This article described how we reached the final checklist and Explanation & Elaboration text and provides information on the decision-making process. We anticipate this paper will help others who may learn from our experiences and may apply this to the development of future guidelines or extensions.

There were several important strengths to our approach. A consensus-driven Delphi methodology, which is recommended when developing healthcare reporting guidelines by the EQUATOR network, was used to develop the extension.[23] We engaged with key stakeholders in trials research and potential end-users of the resultant CONSORT-ROUTINE reporting guideline throughout the development process. The process involved participants from a wide range of scientific disciplines and with diverse experience in conducting trials using different cohorts and routinely collected databases. As with other CONSORT-related guidelines, the inclusion of CONSORT Group members (IB, DM and PR) was intended to ensure consistency in the use of recommended methods in the development, dissemination and implementation of the extension. We recorded high response rates of 74% (92 respondents), 84% (77 respondents) and 81% (62 respondents) in Delphi rounds 1, 2 and 3, respectively. In addition, the number of registered participants and responders is larger than in most Delphi surveys used to develop healthcare reporting guidelines.[8 35 36] Finally, we achieved a high degree of consensus that was consistent across Delphi survey rounds for the majority of the items.

There are also limitations to consider. One is that most participants were academic researchers with primary roles in trials research, and despite our broad engagement efforts, the number of participants from some stakeholder groups was small. One patient was included as a member of the reporting guideline development team, but no patients participated in the Delphi exercise. It is possible that perceptions about the importance of items might have differed across different stakeholder groups that might have favoured the inclusion or exclusion of certain items. Nonetheless, our project group included people from diverse backgrounds with expertise in using different types of data sources, who oversaw the development process to ensure that the checklist was equally applicable to, and representative of, all four types of data sources. A second is that our scoping review was not designed to capture each and every trial conducted using routinely collected data. This was in part because of the lack of accepted specific Medical Subject Headings terms

to identify these studies, or any research using routinely collected data, and the limited number of completed trials and methodological articles on these trial designs. For our purposes, it was not necessary to capture all trials that had been conducted using cohorts or routinely collected data, and we believe that we were able to capture a significant number of important trial reports and methodology papers that served as a basis for the development of our extension. A third is that the CONSORT-ROUTINE group predominantly consisted of members from high-income countries, which might have led to decreased applicability of the checklist for trials conducted in other settings. Finally, as with all reporting guidelines, ours will require re-evaluation and revisions over time to ensure that it is kept up to date with evolving research and knowledge on these trail designs.

## CONCLUSION

CONSORT-ROUTINE has now been developed and can be used to support comprehensive reporting of RCTs conducted using cohorts or routinely collected data. The extension statement contains minimum requirements of reporting that we encourage researchers to report. A baseline assessment of the completeness and reporting of these trial designs is being conducted, and the impact of the extension will be assessed in the coming years. While we anticipate that CONSORT-ROUTINE may need to be updated with the evolution of research methods, we hope the guideline will improve the reporting of RCTs conducted using cohorts and routinely collected data, enhance their interpretability and credibility of their results, improve their reproducibility, indirectly facilitate their robust design and conduct and lead to improved patient care.

**Author affiliations**
[1]Lady Davis Institute for Medical Research, Jewish General Hospital, Montreal, Québec, Canada
[2]Behavioural Science Institute, Clinical Psychology, Radboud University, Nijmegen, Netherlands
[3]National Perinatal Epidemiology Unit Clinical Trials Unit, Nuffield Department of Population Health, University of Oxford, Oxford, UK
[4]Center for Research on Population and Health, Faculty of Health Sciences, American University of Beirut, Ras Beirut, Lebanon
[5]Institute of Applied Health Sciences, School of Medicine, Medical Sciences and Nutrition, University of Aberdeen, Aberdeen, UK
[6]Basel Institute for Clinical Epidemiology and Biostatistics, Department of Clinical Research, University Hospital Basel, University of Basel, Basel, Switzerland
[7]Faculty of Health, Department of Cardiology, Örebro University, Örebro, Sweden
[8]Department of Family Medicine, Western University, London, Ontario, Canada
[9]IC/ES Western, London, Ontario, Canada
[10]Centre for Clinical Trials and Methodology, Barts Institute of Population Health Science, Queen Mary University, London, UK
[11]Department of Psychology, McGill University, Montreal, Quebec, Canada
[12]Faculty of Epidemiology and Population Health, London School of Hygiene and Tropical Medicine, London, UK
[13]Department of Pediatrics and School of Epidemiology and Public Health, University of Ottawa, Ottawa, Ontario, Canada
[14]ICES uOttawa, Ottawa, Ontario, Canada
[15]Children's Hospital of Eastern Ontario Research Institute, Ottawa, Ontario, Canada
[16]Department of Health Research Methods, Evidence, and Impact, McMaster University, Hamilton, Ontario, Canada
[17]Health Services Research Unit, University of Aberdeen, Aberdeen, UK

[18]Library Services, Children's Hospital of Eastern Ontario, Ontario, Ottawa, Canada
[19]Department of Cardiology, Clinical Sciences, Lund University, Lund, Sweden
[20]University Medical Center Utrecht, Utrecht, Netherlands
[21]University of Utrecht, Utrecht, Netherlands
[22]Centre for Journalology, Clinical Epidemiology Program, Ottawa Hospital Research Institute, Ottawa, Ontario, Canada
[23]INSERM, Paris, France
[24]Centre d'Épidémiologie Clinique, Hôpital Hôtel Dieu, Assistance Publique–Hôpitaux de Paris, Paris, France
[25]Faculté de Médecine, Université Paris Descartes, Sorbonne Paris Cité, Paris, France
[26]School of Health and Related Research, University of Sheffield, Sheffield, UK
[27]Department of Psychiatry, Dalhousie University, Halifax, Nova Scotia, Canada
[28]Scleroderma Society of Ontario, Hamilton, Ontario, Canada
[29]Scleroderma Canada, Hamilton, Ontario, Canada
[30]British Medical Journal, London, UK
[31]York Trials Unit, Department of Health Sciences, University of York, York, UK
[32]Neonatal Medicine, School of Public Health, Faculty of Medicine, Imperial College London, London, UK
[33]Nottingham Clinical Trials Unit, University of Nottingham, Nottingham, UK
[34]Departments of Psychiatry; Epidemiology, Biostatistics and Occupational Health; Medicine; and Educational and Counselling Psychology; and Biomedical Ethics Unit, McGill University, Montreal, Quebec, Canada

**Correction notice** This article has been corrected since it was published. Deatils regarding final CONSORT-ROUTINE checklist under section Project phase 5: publication, dissemination and implementation has been corrected.

**Contributors** MI, LK, OF, LGH, MZ, CR, SML, DM, MSam, CG, EJ and BDT were involved in initial phases of study conception, design of the search strategy and development of conceptual frameworks. SJM, KAM, DBR, EIB, LT, MKC, DE, HMV, IB, PR, JN, RU, MSau, JF and DT provided regular feedback on each of these steps. MI wrote the first draft with LK and BDT. All authors made provided critical revisions to the development of this manuscript and approved the final version.

**Funding** The development of CONSORT-ROUTINE was supported by the Canadian Institutes of Health Research (CIHR; PJT156172; PCS-161863), and the UK National Institute of Health Research (NIHR) Clinical Trials Unit Support Funding – Supporting efficient/innovative delivery of NIHR research (Principal Investigator (PI): EJ, co-PI: CG). DBR was supported by a Vanier CIHR Graduate Scholarship; SML was supported by a Wellcome Senior Clinical Fellowship in Science (205039/Z/16/Z); EIB was supported by a New Investigator Award from CIHR, the Canadian Association of Gastroenterology and Crohn's and Colitis Canada, and the Career Enhancement Program of the Canadian Child Health Clinician Scientist Program; RU was supported by the Canada Research Chairs Program (Award #231397); CG was supported by the UK Medical Research Council through a Clinician Scientist Fellowship; and BDT was supported by a Tier 1 Canada Research Chair, all outside of the present work.

**Disclaimer** The views expressed are those of the authors and not necessarily those of the NHS, the NIHR or the Department of Health and Social Care.

**Competing interests** None declared.

**Patient consent for publication** Not required.

**Provenance and peer review** Not commissioned; externally peer reviewed.

**Data availability statement** All data relevant to the study are included in the article or uploaded as supplemental information.

**ORCID iDs**
Ole Fröbert http://orcid.org/0000-0002-5846-345X
Danielle B Rice http://orcid.org/0000-0001-5615-7005
Sinéad M Langan http://orcid.org/0000-0002-7022-7441
David Moher http://orcid.org/0000-0003-2434-4206
Chris Gale http://orcid.org/0000-0003-0707-876X
Brett D Thombs http://orcid.org/0000-0002-5644-8432

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
