## [Reviewer comments · BMJ Open]

This paper was submitted to a another journal from BMJ but declined for publication following peer review. The authors addressed the reviewers' comments and submitted the revised paper to BMJ Open. The paper was subsequently accepted for publication at BMJ Open.

(This paper received two reviews from its previous journal but only one reviewer agreed to published their review.)

ARTICLE DETAILS

TITLE (PROVISIONAL)	Methods and Results Used in the Development of a Consensus-driven Extension to the Consolidated Standards of Reporting Trials (CONSORT) Statement for Trials Conducted Using Cohorts and Routinely Collected Data (CONSORT-ROUTINE)
AUTHORS	Imran, Mahrukh; Kwakkenbos, Linda; McCall, Stephen; McCord, Kimberly; Frobert, Ole; Hemkens, Lars; Zwarenstein, Merrick; Relton, Clare; Rice, Danielle; Langan, Sinead; Benchimol, Eric; Thabane, Lehana; Campbell, Marion; Sampson, Margaret; Erlinge, David; Verkooijen, Helena; Moher, David; Boutron, Isabelle; Ravaud, Philippe; Nicholl, Jon; Uher, Rudolf; Sauve, Maureen; Fletcher, John; Torgerson, David; Gale, Chris; Juszczak, Edmund; Thombs, Brett

VERSION 1 – REVIEW

REVIEWER	Tredinnick-Rowe, John The National Institute for Health Research (NIHR) Applied Research Collaboration (ARC) South West Peninsula
REVIEW RETURNED	10-Jan-2021

GENERAL COMMENTS	Thank you for this updated manuscript It was very interesting to read, and I think there had been lots of effort put into it. Updating reporting standards is always welcome. The paper is potentially useful to many stakeholders The only really specific comment I had was why have patients not included in the stakeholder (Delphi) activities? there needs to be a justification for this. Beyond this I thought that the items are all intelligible and coherent, examples are clear and sensible. The modification to items you have made seems sensible. There are enough modification to warrant publication. You cover a wide range of important factors, ranging from the pragmatic considerations.
--

VERSION 1 – AUTHOR RESPONSE

Reviewer #1		
Thank you for this updated manuscript It was very interesting to read, and I think there had been lots of effort put into it. Updating reporting standards is always welcome. The paper is potentially useful to many stakeholders	We thank Reviewer #1 for this positive comment.	No change.
The only really specific comment I had was why have patients not included in the stakeholder (Delphi) activities? there needs to be a justification for this.	The Delphi activities included people knowledgeable about trial reporting. We do not believe that patients are typically included in this activity, but we do understand that this might be seen as a limitation. Thus, we have noted, “One patient was included as a member of the reporting guideline development team, but no patients participated in the Delphi exercise.”	Page 22.
Beyond this I thought that the items are all intelligible and coherent, examples are clear and sensible. The modification to items you have made seems sensible. There are enough modification to warrant publication.	We thank Reviewer #1 for noting this.	No change.
You cover a wide range of important factors, ranging from the pragmatic considerations.	We thank Reviewer #1 for noting this.	No change.